# Superconformal Line Defects in 3D

Silvia Penati 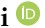

Dipartimento di Fisica, Università degli Studi di Milano-Bicocca, and INFN, Sezione di Milano-Bicocca, Piazza della Scienza 3, 20126 Milano, Italy; silvia.penati@mib.infn.it

**Abstract:** We review the recent progress in the study of line defects in three-dimensional Chern–Simons-matter superconformal field theories, notably the ABJM theory. The first part is focused on kinematical defects, supporting a topological sector of the theory. After reviewing the construction of this sector, we concentrate on the evaluation of topological correlators from the partition function of the mass-deformed ABJM theory and provide evidence on the existence of topological quantum mechanics living on the line. In the second part, we consider the dynamical defects realized as latitude BPS Wilson loops for which an exact evaluation is available in terms of a latitude Matrix Model. We discuss the fundamental relation between these operators, the defect superconformal field theory and bulk physical quantities, such as the Bremsstrahlung function. This relation assigns a privileged role to BPS Wilson operators, which become the meeting point for three exact approaches: localization, integrability and conformal bootstrap.

**Keywords:** Chern–Simons-matter theories; topological operators; Wilson loops

## 1. Introduction

The study of quantum field theories with defects is of crucial importance not only because they can be used to describe real physical systems that exhibit lower dimensional interplays or doping defects, but also because defect theories have proved to be an efficient tool for investigating physical properties of the bulk system itself. This becomes even more efficient when (super)conformal invariance is at work.

Superconformal line defects can be of two types. The first kind of defects are trivial lines viewed as boundary conditions for the functional integral of the theory, or equivalently, as lower dimensional manifolds supporting subsectors of bulk operators. We will refer to them as "kinematical defects". The second kind of defects are extended operators, notably Wilson loops, described by exponentials of integrated defect Lagrangians. We will refer to them as "dynamical defects".

In recent years, a boost in the study of superconformal theories (SCFT) with defects has been achieved, thanks to the use of innovative theoretical tools, such as the AdS/CFT correspondence, integrability, supersymmetric localization, the topological twist and the superconformal bootstrap. We refer to [1] for an introduction to theories with boundaries and defects.

In this review, we will focus on one-dimensional kinematical and dynamical defects in three-dimensional SCFTs that allow for a Lagrangian description in terms of a quiver Chern–Simons theory coupled to a suitable matter sector. Though we can have, in general, $0 \leq \mathcal{N} \leq 8$ supersymmetry, according to the particular matter content and the particular value of the couplings, we will consider as a benchmark the $\mathcal{N} = 6$ $U(N)_k \times U(N)_{-k}$ ABJM theory [2].

Beyond the genuine interest in three-dimensional SCFTs, which arises from the fact that they describe realistic condensed matter systems around quantum critical points, these theories play a central role in formulating the $AdS_4/CFT_3$ correspondence, providing, in principle, a field theory dual formulation of four-dimensional quantum gravity. Though the $AdS_4/CFT_3$ version of the correspondence shares many important features with the

more common AdS$_5$/CFT$_4$ one—it involves supersymmetry and has an underlying integrable structure—it also has peculiar differences, which make the two formulations not a simple replica. The study of three-dimensional SCFTs with or without defects is then of great interest.

After a brief reminder about the ABJM theory, its field content and its symmetries, given in Section 2, the first part of this review will be devoted to the construction of the topological sector of the theory, made by local operators projected on a line, which belongs to the cohomology of a twisted nilpotent supercharge. In Section 3, we will focus on topological correlation functions and present some evidence about their connection with the derivatives of the partition function of the mass-deformed ABJM theory in its matrix representation. This leads to the emergence of a topological quantum mechanics supported by the kinematic defect. Its potential implications for the solvability of the bulk theory, that is, for determining its CFT data (scaling dimensions and OPE coefficients), are briefly addressed.

In Section 4, we move to consider the dynamical defects realized as *latitude* Wilson loops. These are BPS operators corresponding to generalized connections, which include parametric (latitude) couplings to the bosonic and fermionic matter sectors of the theory. These operators play a fundamental role in the development of new exact methods in quantum field theory. First of all, (latitude) Wilson loops are amenable of exact evaluation via localization. On the other hand, as we will review, derivatives of latitude Wilson loops determine the Bremsstrahlung function, which, in turn, can be evaluated using integrability. At the same time, derivatives of latitude Wilson loops give rise to correlation functions of the one-dimensional defect theory, which can be also approached using SCFT techniques, notably Ward identities and the conformal bootstrap. To complete the picture, for some Wilson loops, the dual description in terms of fundamental strings ending on the Wilson contour at the boundary is known. Therefore, BPS Wilson loops are the best playground for testing the consistency among different exact methods—localization, integrability, and conformal bootstrap—and for performing precision tests of the AdS/CFT correspondence. Complementarily, the interplay between SCFT techniques, localization, integrability and holographic techniques makes the study of defect SCFTs very promising.

Section 5 is devoted to some conclusions. A list of interesting open problems that need further investigation is also reported there.

## 2. The ABJM Theory

We begin with a short summary of the field description of the ABJ(M) theories [2,3]. This is a class of three-dimensional $U(N_1)_k \times U(N_2)_{-k}$ quiver theories, whose field content is given by two Chern–Simons gauge vectors, $A_\mu$ and $\hat{A}_\mu$, minimally coupled to $SU(4)$ complex scalars $C_I$, $\bar{C}^I$ and the corresponding fermions $\bar{\psi}^I$, $\psi_I$, $I = 1, \ldots, 4$, all belonging to the (anti)bifundamental representation of the gauge group. The total action is given by

$$S = S_{\text{CS}} + S_{\text{mat}} + S_{\text{pot}}^{\text{bos}} + S_{\text{pot}}^{\text{ferm}} \tag{1}$$

where

$$S_{\text{CS}} = \frac{k}{4\pi i} \int d^3 x \, \varepsilon^{\mu\nu\rho} \left\{ \text{Tr}\left( A_\mu \partial_\nu A_\rho + \frac{2}{3} i A_\mu A_\nu A_\rho \right) - \text{Tr}\left( \hat{A}_\mu \partial_\nu \hat{A}_\rho + \frac{2}{3} i \hat{A}_\mu \hat{A}_\nu \hat{A}_\rho \right) \right\}$$

$$S_{\text{mat}} = \int d^3 x \, \text{Tr}\left[ D_\mu C_I D^\mu \bar{C}^I - i \bar{\Psi}^I \gamma^\mu D_\mu \Psi_I \right] \tag{2}$$

Here $k$ is the Chern–Simons level and $D_\mu$ the covariant derivatives[1]. Matter fields are subject to a non-trivial potential, $(S_{\text{pot}}^{\text{bos}} + S_{\text{pot}}^{\text{ferm}})$ in (1). More precisely, $S_{\text{pot}}^{\text{bos}}$ is a pure scalar sextic potential, whereas $S_{\text{pot}}^{\text{ferm}}$ contains quartic couplings between scalars and fermions. The interested reader can find their explicit expressions in, for instance, [5].

We will primarily focus on the ABJM model with equal gauge ranks, $N_1 = N_2$, though most of the discussion that follows has a simple generalization (with some slight differences) to the more general ABJ theory ($N_1 \neq N_2$).

For a particular choice of the couplings in the potential terms, the theory possesses $\mathcal{N} = 6$ superconformal symmetry. The superconformal algebra is $\mathfrak{osp}(6|4)$, which includes the $\mathfrak{su}(4)$ R-symmetry generators. It can be studied perturbatively in the coupling constant $\lambda = N/k$ for $N \ll k$. In the opposite regime, for $N \gg k^5$ the model is dual to M-theory on $\mathrm{AdS}_4 \times S^7/Z_k$, whereas in the range $k \ll N \ll k^5$, it corresponds to Type IIA string theory on $\mathrm{AdS}_4 \times \mathbb{CP}^3$.

More general quiver Chern–Simons-matter theories have been constructed, which possess $\mathcal{N} = 2, 3, 4, 5$ superconformal symmetry [5–12].

Exact results for the 3D $\mathcal{N} \geq 2$ Chern–Simons-matter theories were obtained, using supersymmetric localization [13,14], which allows to trade the functional integral computing the partition function with a standard matrix integral[2]. For the ABJM theory compactified on the $S^3$ sphere, the partition function is known to be as follows [16]

$$
\mathcal{Z} = \int \prod_{a=1}^{N} d\lambda_a \, e^{i\pi k \lambda_a^2} \prod_{b=1}^{N} d\mu_b \, e^{-i\pi k \mu_b^2} \times \frac{\prod_{a<b}^{N} \sinh^2 \pi(\lambda_a - \lambda_b) \prod_{a<b}^{N} \sinh^2 \pi(\mu_a - \mu_b)}{\prod_{a=1}^{N} \prod_{b=1}^{N} \cosh^2 \pi(\lambda_a - \mu_b)} \tag{3}
$$

Here, the integrals are on the two sets of eigenvalues $\{\lambda_a\}, \{\mu_a\}$ of the Cartan subalgebras of the two $U(N)$ gauge groups. Correlation functions of gauge invariant operators, which preserve the localizing supercharge, can, in principle, be computed exactly from (3) with suitable insertions.

The matrix integral (3) can be checked at weak coupling ($N/k \ll 1$) by matching a genuine perturbative calculation [16], while its expansion at strong coupling reproduces the string holographic prediction [17,18] and provides a non-trivial test of the AdS/CFT correspondence.

The matrix models computing partition functions have been also obtained for the mass deformed ABJM theory [19–23] and for the theory compactified on the squashed sphere [24,25]. We refer to the literature for their explicit expressions.

## 3. Kinematical Defects in the ABJM Theory

Solving the ABJM theory amounts to classifying all the quantum-local and non-local-gauge invariant observables in irreducible representations of the superconformal algebra and computing all their correlation functions. In principle, this can be done by using exact methods, such as supersymmetric localization and the bootstrap approach; however, in practice, computational difficulties can make the program quite challenging. A way to circumvent technical obstacles is to start from investigating suitable subsectors of the theory, where correlators are easier to evaluate, but at the same time, can provide some information on the whole spectrum of the theory.

Remarkable examples are *topological sectors* made up by local operators restricted on lower dimensional subspaces, whose correlation functions are space-time independent. These sectors can be constructed whenever the theory, once reduced to the given subspace, possesses enough supersymmetry to allow for a topological twist, as originally introduced by Witten [13]. The original Lorentz group on the lower dimensional subspace is traded for a twisted Lorentz group, whose generators are given by linear combinations of the original Lorentz and R-symmetry ones. The new generators turn out to be $Q$-exact in the cohomology of a spin-zero cohomological supercharge $Q$, given by a linear combination of the original supersymmetry and superconformal charges. As a consequence, correlators of local operators in the $Q$-cohomology are independent of the metric and the space-time coordinates. Representatives of the $Q$-cohomology classes are then dubbed *topological operators*.

Here, we review how this procedure can be applied to construct a one-dimensional topological subsector of the ABJM theory [26].

### 3.1. The Topological Line of ABJM

The topological sector of the ABJM theory is constructed from local gauge-invariant operators, restricted to live on a kinematic defect, that is, a trivial straight line. For the scope of constructing it explicitly, in the Euclidean three-dimensional space, we consider a line parallel to the $x^3$-direction, parametrized as $x^\mu(s) = (0, 0, s)$, with $s \in (-\infty, +\infty)$ being its proper time.

Fixing this kinematic defect breaks the original $\mathcal{N} = 6$ superconformal algebra $\mathfrak{osp}(6|4)$ to $\mathfrak{su}(1,1|3) \oplus \mathfrak{u}(1)_b$, whose generators are given by[3]

$$
\begin{aligned}
&\text{1D conformal algebra } \mathfrak{sl}(2): \quad &&(P, K, D) \\
&\text{R} - \text{symmetry } \mathfrak{su}(3): \quad &&R_a{}^b \quad &&a, b = 1, 2, 3 \\
&\text{Super(conformal) charges}: \quad &&Q^a, \bar{Q}_a, S^a, \bar{S}_a \quad &&a, b = 1, 2, 3 \\
&\mathfrak{u}(1)_m: \quad &&M = 3iM_{12} - 2J_1{}^1 \\
&\mathfrak{u}(1)_b: \quad &&B = M_{12} + 2iJ_1{}^1
\end{aligned}
\tag{4}
$$

here, $P$ is the translation operator along the line, while $K$ and $D$ generate special conformal transformations and dilations, respectively. The super(conformal) charges satisfy the hermitian conjugation rules $(S^a)^\dagger = \bar{Q}_a$, $(\bar{S}_a)^\dagger = Q^a$.

Unitary irreducible representations (UIR) of the $\mathfrak{su}(1,1|3)$ algebra are labeled by four quantum numbers, $[\Delta, m, j_1, j_2]$, where $\Delta$ is the conformal weight, $m$ the $\mathfrak{u}(1)_m$ charge, while $j_1, j_2$ are the eigenvalues of the two $\mathfrak{su}(3)$ Cartan matrices. An exhaustive classification of UIRs can be found in [27] (see also [26,28]).

The elementary matter fields listed in Section 2 can be reorganized according to $SU(3)$ representations as follows

$$
C_I = (Z, Y_a) \qquad \bar{C}^I = (\bar{Z}, \bar{Y}^a) \qquad \psi_I = (\psi, \chi_a) \qquad \bar{\psi}^I = (\bar{\psi}, \bar{\chi}^a) \qquad a = 1, 2, 3 \tag{5}
$$

where $Y_a(\bar{Y}^a), \chi_a(\bar{\chi}^a)$ belong to the $3(\bar{3})$ of $SU(3)$, while $Z, \bar{Z}, \psi, \bar{\psi}$ are $SU(3)$-singlets. They provide the building blocks for the field realization of UIRs in terms of local, gauge invariant operators.

**The topological twist.** In order to perform the topological twist, inside the complexified $\mathfrak{su}(3)_\mathbb{C}$ R-symmetry algebra, we select the $\mathfrak{su}(1,1)(\simeq \mathfrak{sl}(2))$ subalgebra generated by

$$
\left( iR_3{}^1, \ iR_1{}^3, \ \frac{R_1{}^1 - R_3{}^3}{2} \right) \equiv (\mathcal{R}_+, \mathcal{R}_-, \mathcal{R}_0) \tag{6}
$$

With respect to this subalgebra, the supercharges in (4) split into two doublets $(Q^1, Q^3)$ and $(S^1, S^3)$, and their Hermitian conjugates $(\bar{Q}_1, \bar{Q}_3)$, $(\bar{S}_1, \bar{S}_3)$, which transform in the fundamental of $\mathfrak{su}(1,1)$ and have $\mathfrak{u}(1)$ charges $1/6$ and $-1/6$, respectively. The remaining supercharges $Q^2, S^2$ $(\bar{Q}_2, \bar{S}_2)$ are instead singlets with $U(1)$ charges $-1/3$ $(1/3)$.

The topological twist is now performed by taking a suitable diagonal sum of the original spacetime conformal algebra defined in (4) with the $\mathfrak{su}(1,1)$ given in (6). The new generators

$$
\hat{L}_+ = P + \mathcal{R}_+ \qquad \hat{L}_- = K + \mathcal{R}_- \qquad \hat{L}_0 = D + \mathcal{R}_0 \tag{7}
$$

satisfy a $\widehat{\mathfrak{su}}(1,1)$ conformal algebra, the "twisted" algebra, which, together with the two nilpotent supercharges

$$
\mathcal{Q}_\pm = \frac{1}{\sqrt{2}} \left( Q^3 + iS^1 \pm (\bar{S}_3 + i\bar{Q}_1) \right), \qquad \mathcal{Q}_+^2 = \mathcal{Q}_-^2 = 0 \tag{8}
$$

form a superalgebra with central extension $\mathcal{Z} = \frac{1}{4}\{\mathcal{Q}_-, \mathcal{Q}_+\}$.

The remarkable fact is that the $\widehat{\mathfrak{su}}(1,1)$ generators are $\mathcal{Q}_\pm$-exact. In fact, it is easy to check that

$$\hat{L}_+ = \{\mathcal{Q}_+, \tilde{Q}_+\} = \{\mathcal{Q}_-, \tilde{Q}_-\} \qquad \hat{L}_- = \{\mathcal{Q}_+, \tilde{S}_+\} = \{\mathcal{Q}_-, \tilde{S}_-\}$$
$$\hat{L}_0 = \frac{1}{2}\left\{\mathcal{Q}_+, \mathcal{Q}_+^\dagger\right\} = \frac{1}{2}\left\{\mathcal{Q}_-, \mathcal{Q}_-^\dagger\right\} \tag{9}$$

where $\tilde{Q}_\pm = \frac{1}{\sqrt{2}}(\bar{Q}_3 \mp iQ^1)$ and $\tilde{S}_\pm = \frac{1}{\sqrt{2}}(-i\bar{S}_1 \pm S^3)$.

**The cohomology.** The cohomology of the $\mathcal{Q}_\pm$ charges is built by the set of local, gauge invariant operators $\mathcal{O}(s)$ living on the line and satisfying[4]:

$$[Q, \mathcal{O}(s)]_\pm = 0, \qquad \mathcal{O}(s) \neq [Q, \mathcal{O}'(s)]_\mp \tag{10}$$

We can solve the cohomological equations at the origin ($s = 0$) and then move the operator along the line by acting with the twisted translation operator $\hat{L}_+$, according to

$$\mathcal{O}(s) \equiv e^{-s\hat{L}_+}\,\mathcal{O}(0)\,e^{s\hat{L}_+} \tag{11}$$

In fact, since $\hat{L}_+$ is $\mathcal{Q}$-exact (see Equation (9)), translating the operator away from the origin does not affect the cohomology.

Similarly, since $\hat{L}_0$ and $\mathcal{Z}$ generators are $\mathcal{Q}$-exact, they act trivially within each cohomological class. Therefore, representatives of the $\mathcal{Q}$-classes belong necessarily to the zero eigenspaces of $\hat{L}_0$ and $\mathcal{Z}$ [29]. Vice versa, in a unitary representation, any element of the kernel of $\hat{L}_0$ must be annihilated by $\mathcal{Q}$. Now, observing that the highest weight of an irreducible representation of the $\mathfrak{su}(1,1|3)$ superalgebra with quantum numbers $[\Delta, m, j_1, j_2]$ is an eigenvector of $\hat{L}_0$ and $\mathcal{Z}$ with eigenvalues $\hat{l}_0 = \Delta - \frac{j_2 + j_1}{2}$ and $z = \frac{1}{3}\left(m - \frac{j_2 - j_1}{2}\right)$, respectively, it follows that the $\mathcal{Q}$-cohomology classes are in one-to-one correspondence with $[\Delta, m, j_1, j_2]$ representations satisfying

$$\Delta = \frac{j_2 + j_1}{2}, \qquad m = \frac{j_2 - j_1}{2} \tag{12}$$

Scanning all the irreducible representations of $\mathfrak{su}(1,1|3)$ [27,28], we find that constraints (12) are always satisfied by the superconformal primaries of the following short multiplets[5]

$$\mathcal{B}_{\frac{j_2-j_1}{2};j_1,j_2}^{\frac{1}{6},\frac{1}{6}}, \qquad \mathcal{B}_{\frac{j_2-j_1}{2};j_1,j_2}^{\frac{1}{6},0}, \qquad \mathcal{B}_{\frac{j_2-j_1}{2};j_1,j_2}^{0,\frac{1}{6}} \tag{13}$$

for generic values of $j_1, j_2$. For $j_1 = 0$ and/or $j_2 = 0$, the multiplets become shorter and enhance their degree of supersymmetry.

**The field realization.** Knowing the Lagrangian description of the ABJM theory, we can realize topological operators as composite operators built out of the fundamental matter fields restricted to live on the one-dimensional kinematical defect. Taking into account their explicit quantum numbers assignment (see, for instance, Tables 2 and 3 of [26]) it is easy to see that the operator

$$\mathcal{O}_n(0) = \text{Tr}(Y_1 \bar{Y}^3)^n \tag{14}$$

defined at the origin satisfies the cohomological constraints (39,12) with $[\Delta, m, j_1, j_2] = [n, 0, n, n]$. It is the superconformal primary of the $\mathcal{B}_{0;n,n}^{\frac{1}{6},\frac{1}{6}}$ short multiplet. Therefore, it is 1/6 BPS on the line.

For simplicity, we will focus on $\mathcal{O}_1(0) = \text{Tr}(Y_1 \bar{Y}^3)$, which we rename $\mathcal{O}$. Applying twisted translation (11), from direct inspection, we obtain the operator at position $(0, 0, s)$ on the line, as follows

$$\mathcal{O}(s) = \text{Tr}(Y_a(s)\bar{Y}^b(s))\,\bar{u}^a(s)\,v_b(s)\,, \qquad \text{with} \quad \bar{u}^a(s) = (1, 0, is) \qquad v_a(s) = (-is, 0, 1) \tag{15}$$

or more explicitly

$$\mathcal{O}(s) = \text{Tr}(Y_1 \bar{Y}^3) - is\,\text{Tr}(Y_1 \bar{Y}^1) + is\,\text{Tr}(Y_3 \bar{Y}^3) + s^2\,\text{Tr}(Y_3 \bar{Y}^1) \tag{16}$$

We note that the operator is given by a position-dependent linear combination of superconformal primaries. This is reminiscent of what happens in 4D $\mathcal{N} = 4$ SYM theory [30] and in $\mathcal{N} = 4, 8$ three-dimensional theories [31].

It is important to stress that the twisted translation leads to (15) when we use indifferently either the $\mathcal{Q}_+$ or the $\mathcal{Q}_-$ cohomology. One could be tempted to conclude that things should work nicely also by considering the cohomology of an arbitrary linear combination $(a_+ \mathcal{Q}_+ + a_- \mathcal{Q}_-)$ with complex numbers $a_\pm$. However, this is not true in general. As explained in [32], good cohomological charges are only the ones corresponding to linear combinations of the following form

$$\mathcal{Q}_\beta = \frac{1}{\sqrt{2}}\left(Q^3 + iS^1 + e^{i\beta}(\bar{S}_3 + i\bar{Q}_1)\right)\,, \qquad \beta \in \mathbb{R} \tag{17}$$

all satisfying $\mathcal{Q}_\beta^2 = 0$. In conclusion, we have a one-parameter family of cohomological supercharges that can be used to perform the topological twist on the line.

### 3.2. The 1D Topological Correlators

We are interested in evaluating correlation functions of the topological operators defined in the previous section. Focusing on $\mathcal{O}$, we study the generic $n$-point function

$$\langle \mathcal{O}(s_1) \cdots \mathcal{O}(s_n) \rangle \equiv \int \mathcal{O}(s_1) \cdots \mathcal{O}(s_n)\, e^{-S} \tag{18}$$

As anticipated in the previous discussion, these correlators are expected to be, in general, non-vanishing and position independent. In fact, since from (11) it follows that $\partial_s \mathcal{O}(s) = -[\hat{L}_+, \mathcal{O}(s)]$ and $\hat{L}_+$ is $\mathcal{Q}$-exact, we obtain

$$\partial_{s_j}\langle \mathcal{O}(s_1) \cdots \mathcal{O}(s_j) \cdots \mathcal{O}(s_n) \rangle = \langle \left\{ \mathcal{Q}, \mathcal{O}(s_1) \cdots [\tilde{\mathcal{Q}}, \mathcal{O}(s_j)] \cdots \mathcal{O}(s_n) \right\} \rangle = 0\,, \ \forall\, j = 1, \cdots, n \tag{19}$$

Moreover, since $\mathcal{O}(s)$ is $1/6$ BPS, we expect these correlators to acquire at most *finite* quantum corrections.

An important observation is now in order. What makes $\mathcal{O}$ special inside the class (14) of topological operators is that it coincides with one of the scalar chiral superprimaries (in $SU(4)$ notation)

$$\mathcal{O}_I{}^J(\vec{x}) = \text{Tr}(C_I(\vec{x})\bar{C}^J(\vec{x})) - \frac{1}{4}\delta_I{}^J\,\text{Tr}(C_K(\vec{x})\bar{C}^K(\vec{x})) \tag{20}$$

which seat in the stress–energy tensor multiplet. In fact, using decomposition (5), it is easy to see that $\mathcal{O}$ is nothing but $\mathcal{O}_2{}^4$ in (20) localized on the line at the origin. Therefore, we expect its correlation functions (18) to carry some information about the correlation functions of the stress–energy tensor, itself. In particular, its two-point function can be used to evaluate the central charge $c_T$ of the ABJM theory. In fact, if we project the general identity

$$\langle \mathcal{O}_I{}^J(\vec{x})\, \mathcal{O}_K{}^L(\vec{0}) \rangle = \frac{c_T}{16}\left(\delta_I^L \delta_K^J - \frac{1}{4}\delta_I^J \delta_K^L\right)\frac{1}{16\pi^2 \vec{x}^2} \tag{21}$$

on the line by setting $\vec{x} = (0,0,s)$ and multiplying by the polarization vectors $\bar{U}^a(s) = (0, \bar{u}^a(s))$, $V_a(s) = (0, v_a(s))$ obtained by promoting the ones in (15) to $SU(4)$ notation, we obtain

$$c_T = -256\,\pi^2\,\langle \mathcal{O}(s)\mathcal{O}(0)\rangle \tag{22}$$

This is an example on how we can extract information on the bulk theory from the kinematical defect.

**The perturbative result.** The perturbative evaluation of correlation functions relies on the expansion of the Euclidean path integral (18) in powers of the ABJM coupling constant $N/k$.

As anticipated, we expect the correlators to be constant, at most depending on the order of the operators along the line. At tree level, this happens since the worldline dependence at the denominator encoded in the propagators is canceled by an analogous numerator coming from the contraction of the polarization vectors [26]. The evaluation of loop contributions reveals that there are no one-loop corrections, whereas the two-point function at two loops reads [26]

$$\langle \mathcal{O}(s)\mathcal{O}(0)\rangle^{(2)} = -\frac{N^2}{(4\pi)^2}\left(1 - \frac{\pi^2}{3k^2}(N^2 - 1)\right) \tag{23}$$

From this result, exploiting (22), we can read the two-loop result for the central charge of ABJM theory, which turns out to be

$$c_T = 16N^2\left(1 - \frac{\pi^2}{3k^2}(N^2 - 1) + O\left(\frac{1}{k^3}\right)\right) \tag{24}$$

Higher order contributions are, in principle, computable, but the evaluation of Feynman integrals becomes more and more challenging. It is then convenient to rely on different approaches for computing the two-point function, as we now describe.

**The correlators from the matrix model.** As already mentioned, the partition function on $S^3$ for the ABJM theory is known exactly from localization (see Equation (3)). In principle, the same technique can be applied to compute correlators (18). Under compactification on the three spheres, the infinite line in $\mathbb{R}^3$ gets mapped to the great circle $S^1 \subset S^3$. Accordingly, the topological operator (15) gets mapped into its spherical version $\mathcal{O}(\varphi)$, which is nothing but the operator evaluated on the great circle parametrized by $0 \leq \varphi \leq 2\pi$, contracted with the polarization vectors on the sphere, $\bar{u}_S^a = (\cos\frac{\varphi}{2}, 0, \sin\frac{\varphi}{2})$, $v_{S\,a} = (-\sin\frac{\varphi}{2}, 0, \cos\frac{\varphi}{2})$ [26]. Superconformal invariance ensures that

$$\langle \mathcal{O}(s_1)\cdots\mathcal{O}(s_n)\rangle_{\mathbb{R}^3} = \langle \mathcal{O}(\varphi_1)\cdots\mathcal{O}(\varphi_n)\rangle_{S^3} \tag{25}$$

It is important to stress that the topological operator $\mathcal{O}$ is no longer invariant under the action of the localizing supercharge used in [16] to obtain the matrix model (3). The only possible supercharges that can be used in the localization procedure are the cohomological supercharges $\mathcal{Q}_\beta$ in Equation (17), which are symmetries of the theory and kill the topological operators. Redoing the localization with these supercharge requires to first bring the cohomological charge off-shell, finding a convenient $Q$-exact term to deform the action and then localize the integrand.

The problem was first attacked in [33] for the case of $\mathcal{N} = 4$ SCFTs described by Yang–Mills vector multiplets suitably coupled to hypermultiplets. Using the nilpotent supercharge $Q$, which features the one-dimensional topological sector of the Higgs branch, one obtains a different but equivalent matrix model for the $\mathcal{N} = 4$ partition function $\mathcal{Z}[S^3]$, which can be interpreted as coming from the gauge sector minimally coupled to a one-dimensional Gaussian model localized on the great circle $S^{16}$. Remarkably, this one-dimensional factor coincides exactly with the contribution from the one-dimensional topological sector defined by the $Q$-cohomology. In fact, it is described by the action

$$S_\sigma = -4\pi r \int_{-\pi}^{\pi} d\varphi \; \bar{\mathcal{J}}_a (\partial_\varphi + \sigma) \mathcal{J}^a \tag{26}$$

where $\mathcal{J}^a$ are dimension-1 topological operators[7], $\sigma$ is the scalar in the three-dimensional vector multiplet, and $r$ is the radius of the sphere. It follows that correlators can be computed with the ordinary prescription

$$\langle \mathcal{J}^{a_1}(s_1) \cdots \mathcal{J}^{a_n}(s_n) \rangle \sim \int [D\mathcal{J} D\bar{\mathcal{J}}] \, e^{-S_\sigma} \, \mathcal{J}^{a_1}(s_1) \cdots \mathcal{J}^{a_n}(s_n) \tag{27}$$

for a one-dimensional quantum mechanics.

A few comments are now in order. First of all, if we remove the operator insertions the matrix model reduces to the one in Equation (3) evaluating the partition function [33], consistent with the fact that the result for the partition function must be independent of the choice of the localizing supercharge. Second, it can be proved that the SYM action is $Q$-exact, with respect to the cohomological supercharge. Therefore, in three-dimensional $\mathcal{N} = 4$ SCFT theories with a Yang–Mills-type action, correlators (27) are expected to be independent of the coupling constant.

This results can be easily generalized to non-conformal theories obtained by deforming the original SCFT with mass parameters $m^a$. The matrix model computing the partition function of the mass deformed theory on $S^3$ is known in the large $N$ limit [22,23] and exactly [36]. On the other hand, in the alternative derivation described above this deformation is equivalent to add mass terms of the form $-4\pi r^2 m^a \int_{-\pi}^{\pi} d\tau \, \mathcal{J}^a(\tau)$ to the one-dimensional Gaussian model (26) [33]. It follows that taking derivatives of the matrix model on $S^3$ respect to the mass parameters $m^a$ is equivalent to bring down factors $-4\pi r^2 \int_{-\pi}^{\pi} d\tau \, \mathcal{J}^a(\tau)$ inside the Gaussian one-dimensional functional, so obtaining integrated correlation functions of topologically twisted operators living on the great circle. Precisely, the following remarkable identity holds [37,38]

$$\left\langle \int_{-\pi}^{\pi} d\tau_1 \dots \int_{-\pi}^{\pi} d\tau_n \, \mathcal{J}^{a_1}(\tau_1) \dots \mathcal{J}^{a_n}(\tau_n) \right\rangle = \frac{1}{(4\pi r^2)^n} \frac{1}{\mathcal{Z}} \frac{\partial^n}{\partial m^{a_1} \dots \partial m^{a_n}} \mathcal{Z}[S^3, m^a] \Big|_{m^a = 0} \tag{28}$$

In particular, since the topological correlators are position independent, the integrals on the l.h.s. can be trivially performed, leading to a constant factor $(2\pi)^n$ times the correlator. Therefore, identity (28) provides an exact prescription for computing correlators in the one-dimensional topological sector in terms of the derivatives of the deformed matrix model of the three-dimensional theory. Read in the opposite direction, this allows to reconstruct the exact partition function of the three-dimensional theory on the sphere once we have solved the one-dimensional topological theory, i.e., we know exactly all its correlators.

This procedure can be generalized to $\mathcal{N} = 8$ SCFTs [37], being these theories special cases of $\mathcal{N} = 4$ SCFTs with an extra $\mathfrak{so}(4)$ flavor symmetry. In this case, the topological sector is constructed from the three-dimensional operators in (20), which belong to the $\mathcal{N} = 8$ stress-energy tensor multiplet. Ward identities then relate topological correlators to the ones of the stress-energy tensor in a particular kinematic configuration. In particular, the topological two-point function is related to the central charge as in (22). On the other hand, the topological two-point function can be computed, using prescription (28). Putting everything together, it then follows that $c_T$ is related to the second derivative of the mass-deformed partition function, according to

$$c_T = -\frac{64}{\pi^2} \frac{d^2}{dm^2} \log \mathcal{Z}[S^3, m] \Big|_{m=0} \tag{29}$$

This coincides with the relation found in [39] for $\mathcal{N} \geq 2$ SCFTs, using an alternative approach. This is a non-trivial check of identity (28) for the $\mathcal{N} = 8$ case.

For $\mathcal{N} = 8$ and $\mathcal{N} = 4$ SCFTs, the topological sector has played a notable role in performing a precision study of the theories through conformal bootstrap, allowing to compute

exactly some OPE data and constraining "islands" in the parameter space [31,37,40,41]. At the same time, it was instrumental in fixing contributions to the scattering amplitudes of super-gravitons in M-theory in 11 dimensions [42].

For $\mathcal{N} = 6$ ABJ(M) theory, the topological sector was considered in connection with string theory amplitudes in $\text{AdS}_4 \times \text{CP}^3$ [38]. However, for this case, a direct derivation of identity (28) is not available since the absence of a $\mathcal{N} = 4$ SYM mirror theory and the presence of Chern–Simons terms somehow preclude a direct derivation of a one-dimensional action for the topological sector. It is then necessary to provide indirect evidence of identity (28) through the use of alternative approaches.

A first piece of evidence was recently given in [26] by a perturbative evaluation of identity (29) for the ABJM theory. In fact, referring to the topological operator $\mathcal{O}$ in (15), it was shown there that the two-loop result (24) for the central charge coming from a genuine two-loop evaluation of $\langle \mathcal{O}(s)\mathcal{O}(0) \rangle$ matches exactly the second derivative of the mass-deformed matrix model of ABJ(M) on $S^3$ [19–21]

$$\mathcal{Z} = \frac{1}{(N!)^2} \int d\lambda \, d\mu \, \frac{e^{i\pi k \sum_i (\lambda_i^2 - \mu_i^2)} \prod_{i<j} 16 \sinh^2 \left[\pi(\lambda_i - \lambda_j)\right] \sinh^2 \left[\pi(\mu_i - \mu_j)\right]}{\prod_{i,j} 4 \cosh \left[\pi(\lambda_i - \mu_j) + \frac{\pi m_+}{2}\right] \cosh \left[\pi(\lambda_i - \mu_j) + \frac{\pi m_-}{2}\right]} \tag{30}$$

respect to $m_+$ or equivalently $m_-$, where $m_{\pm}$ are the mass assignments of the fundamental scalars $(Z, Y_a) \to (m_+, -m_+, m_-, -m_-)$ in the mass-deformed ABJM theory.

As a last observation, we note that in $\mathcal{N} = 6, 8$ Chern–Simons-matter theories, the Chern–Simons Lagrangian is not $\mathcal{Q}$-exact, regardless of the $\mathcal{Q}$ supercharge that we use to localize the functional integral that computes correlators. Therefore, topological correlators are expected to depend in general on the coupling constant of the theory. The perturbative result given in Equation (23) confirms this expectation.

## 4. Dynamical Defects: BPS Wilson Loops

A notable class of dynamical one-dimensional defects in $U(N) \times U(N)$ ABJM theory is made by the supersymmetric/BPS Wilson loops. These are non-local, gauge invariant operators of the following form

$$W = \text{Tr} P e^{-i \int_\Gamma \mathcal{L}} \qquad \hat{W} = \text{Tr} P e^{-i \int_\Gamma \hat{\mathcal{L}}} \tag{31}$$

where $\mathcal{L}, \hat{\mathcal{L}}$ are generalized connections for the two gauge groups respectively, whose structure is detailed below, and $\Gamma$ is an open or closed one-dimensional contour[8].

For a suitable choice of $\mathcal{L}, \hat{\mathcal{L}}$ and the shape of $\Gamma$, these operators may preserve a fraction of the supersymmetry charges of the theory. This protects them from acquiring divergent contributions at quantum level. Nevertheless, their vacuum expectation value is, in general, a (finite) non-trivial function of the coupling constant, which interpolates between the weak and the strong regimes. Therefore, they represent a powerful tool for proving the AdS/CFT correspondence and a natural playground for testing non-perturbative methods.

A one-dimensional defect SCFT can be defined on a Wilson line/loop by restricting subsets of ABJM local operators to live on the Wilson line. The one-dimensional observables are correlation functions of these local operators computed on the non-trivial vacuum dressed with the Wilson line. Precisely, for a generic operator $\mathcal{O}$ on the infinite straight line we define correlators as

$$\langle\langle \text{Tr} \, \mathcal{O}(s_n)\mathcal{O}(s_{n-1}) \cdots \mathcal{O}(s_1) \rangle\rangle =$$
$$\frac{\langle \text{Tr} W_{s_n, +\infty} \mathcal{O}(s_n) W_{s_{n-1}, s_n} \mathcal{O}(s_{n-1}) \cdots W_{s_1, s_2} \mathcal{O}(s_1) W_{-\infty, s_1} \rangle}{\langle \text{Tr} W_{-\infty, +\infty} \rangle} \tag{32}$$

where we have used the notation $W_{a,b} \equiv P e^{-i \int_a^b \mathcal{L}}$.

An efficient way to insert local operators along the Wilson loop is by applying a broken symmetry generator to the operator itself [44]. For instance, applying the generators of the

transverse translations, we obtain correlators of operators belonging to the displacement multiplet [45]. Alternatively, along the lines described above for the topological line, one can consider the matrix model, computing the vacuum expectation value of parametric Wilson operators, and take derivatives respect to the parameters. We will comment further on this point in Section 4.5. For the time being, we focus on the classification and the quantum properties of the Wilson loops in the ABJM theory and their relation with other important physical quantities.

*4.1. The General Classification*

Ordinary gauge invariant Wilson operators $W = \mathrm{Tr} P e^{-i \int_\Gamma dx^\mu A_\mu}$, $\hat{W} = \mathrm{Tr} P e^{-i \int_\Gamma dx^\mu \hat{A}_\mu}$ break all the supersymmetries of the ABJM theory. However, generalizing the connections $A_\mu, \hat{A}_\mu$ to include also couplings with the matter sector may enhance a fraction of the supersymmetry. Based on the dimensional and group representation arguments, it is easy to see that in three dimensions, we can, in principle, include couplings to bilinear scalars (dimension-one operators in the adjoint of the gauge group) and fermions (dimension-one fields in the (anti)bifundamental).

"Bosonic" Wilson operators that include only couplings to scalar matter was originally proposed in [7] and further elaborated in [46–48]. They correspond to generalized connections for the two $U(N)$ gauge groups, of the following form

$$\mathcal{L}_B = A_\mu \dot{x}^\mu - \frac{2\pi i}{k} |\dot{x}| \mathcal{M}_J^I C_I \bar{C}^J \,, \qquad \hat{\mathcal{L}}_B = \hat{A}_\mu \dot{x}^\mu - \frac{2\pi i}{k} |\dot{x}| \mathcal{M}_J^I \bar{C}^J C_I \qquad (33)$$

where $\mathcal{M}$ is a constant matrix featuring the coupling to scalars. For $\mathcal{M} = \mathrm{diag}(1, 1, -1, -1)$ and choosing the contour to be the infinite straight line or the great circle the two Wilson operators $W_B, \hat{W}_B$ become 1/6 BPS. These operators have a dual description in terms of fundamental type IIA strings ending on the Wilson contour at the $\mathrm{AdS}_4$ boundary and smeared along a $\mathbb{CP}^1$ inside $\mathbb{CP}^3$ [47–51].

As proposed in [52], enhancement of supersymmetry can be obtained by promoting the generalized connection to be an even supermatrix belonging to the $U(N|N)$ supergroup, which includes also fermionic couplings in the off-diagonal blocks. Precisely, it has the following form

$$\mathcal{L}_F = \begin{pmatrix} \mathcal{A} & -i\sqrt{\frac{2\pi}{k}} |\dot{x}| \eta_I \bar{\psi}^I \\ -i\sqrt{\frac{2\pi}{k}} |\dot{x}| \psi_I \bar{\eta}^I & \hat{\mathcal{A}} \end{pmatrix} \quad \text{with} \quad \begin{cases} \mathcal{A} \equiv A_\mu \dot{x}^\mu - \frac{2\pi i}{k} |\dot{x}| \tilde{\mathcal{M}}_J{}^I C_I \bar{C}^J \\ \hat{\mathcal{A}} \equiv \hat{A}_\mu \dot{x}^\mu - \frac{2\pi i}{k} |\dot{x}| \tilde{\mathcal{M}}_J{}^I \bar{C}^J C_I \end{cases} \qquad (34)$$

where $\eta_I, \bar{\eta}^I$ are commuting spinors which drive the coupling to fermions. Choosing the contour to be the straight line or the great circle, $\tilde{\mathcal{M}} = \mathrm{diag}(-1, 1, 1, 1)$ and suitably fixing the couplings to fermions this operator turns out to be 1/2 BPS [52]. It is dual to the 1/2 BPS fundamental string ending on the Wilson contour at the $\mathrm{AdS}_4$ boundary and localized in $\mathbb{CP}^3$. Because of the inclusion of fermions, it is sometimes called the "fermionic" Wilson operator, here $W_F$. According to the general prescription introduced in [53,54], its expression can be derived by the Higgsing $U(N+1) \times U(N+1)$ ABJM theory down to $U(N) \times U(N)$ via the assignment of a non-vanishing vacuum expectation value (vev) to one of the scalars [55].

More general fermionic operators can be defined by allowing the couplings to the matter sector to depend on some arbitrary parameter [56]. According to the general classification given in [57] (see also [58] for a short summary), there exist four classes of fermionic 1/6 BPS Wilson operators $W_F^I, W_F^{II}, W_F^{III}, W_F^{IV}$, which differ for the specific couplings to scalars and fermions and include the 1/2 BPS operator $W_F$ for a special choice of the couplings. They all preserve the same spectrum of supercharges and are

cohomologically equivalent to a bosonic 1/6 BPS Wilson loop $W_{bos}$ corresponding to the superconnection

$$\mathcal{L}_{bos} = \begin{pmatrix} \mathcal{L}_B & 0 \\ 0 & \hat{\mathcal{L}}_B \end{pmatrix} \tag{35}$$

with $\mathcal{L}_B, \hat{\mathcal{L}}_B$ given in (33). In other words,

$$W_F^{I,II,III,IV} = W_{bos} + \mathcal{Q}-\text{exact term} \tag{36}$$

where $\mathcal{Q}$ is a linear combination of supercharges preserved by all the operators. For a particular choice of the parameters, this states the cohomological equivalence between the 1/2 BPS operator $W_F$ and the 1/6 BPS $W_{bos}$, first discovered in [52].

### 4.2. The "Latitude" Wilson Loops

Another set of bosonic and fermionic Wilson operators were introduced in [59,60]. These are obtained from the original operators (33) and (34) evaluated on the great circle by rotating the internal scalar couplings by an angle $\alpha$ and/or deforming the contour to a latitude circle on $S^3$ featured by a latitude angle $\theta$[9]. Though the two deformations are in principle independent, the general expression of the latitude Wilson loops turns out to depend only on the effective parameter $\nu = \sin 2\alpha \cos \theta$ [60].

The general structure of the latitude bosonic connections are still as in (33), but with modified coupling given by

$$\mathcal{M}_J^I(\nu, \tau) = \begin{pmatrix} -\nu & e^{-i\tau}\sqrt{1-\nu^2} & 0 & 0 \\ e^{i\tau}\sqrt{1-\nu^2} & \nu & 0 & 0 \\ 0 & 0 & -1 & 0 \\ 0 & 0 & 0 & 1 \end{pmatrix} \tag{37}$$

For generic values of $\nu \in [0,1]$, these operators are 1/12 BPS, that is, they preserve two independent linear combinations of the original $\mathcal{N} = 6$ supercharges, $\mathcal{Q}_1(\nu)$ and $\mathcal{Q}_2(\nu)$, whose coefficients depend explicitly on $\nu$ [60]. For the special value $\nu = 1$, the matrix $\mathcal{M}$ reduces to $\text{diag}(\mathcal{L}, \hat{\mathcal{L}})$, with $\mathcal{L}, \hat{\mathcal{L}}$ given in (33), and the supersymmetry is enhanced to 1/6 BPS, as discussed in the previous subsection.

Similarly, the latitude fermionic operator is still given in (34), but with the more general couplings

$$\tilde{\mathcal{M}}_I^J(\nu, \tau) = \begin{pmatrix} -\nu & e^{-i\tau}\sqrt{1-\nu^2} & 0 & 0 \\ e^{i\tau}\sqrt{1-\nu^2} & \nu & 0 & 0 \\ 0 & 0 & 1 & 0 \\ 0 & 0 & 0 & 1 \end{pmatrix}, \quad \eta_I^\alpha(\nu, \tau) = \frac{e^{\frac{i\nu\tau}{2}}}{\sqrt{2}} \begin{pmatrix} \sqrt{1+\nu} \\ -\sqrt{1-\nu}e^{i\tau} \\ 0 \\ 0 \end{pmatrix}_I (1, -ie^{-i\tau})^\alpha$$

$$\bar{\eta}_\alpha^I = i(\eta_I^\alpha)^\dagger \tag{38}$$

For generic $\nu$, this operator is 1/6 BPS, while for $\nu = 1$, it enhances to the 1/2 BPS described by superconnection (34)[10].

Both the operators have a smooth limit for $\nu \to 0$, where they give rise to Zarembo-like Wilson loops [61].

Fermionic latitude operators are dual to 1/6 BPS string configurations in $\text{AdS}_4 \times \mathbb{CP}^3$ with the endpoints describing a circle inside $\mathbb{CP}^3$ [62]. The latitude parameter corresponds to a constant boundary condition for one of the $\mathbb{CP}^3$ angular variables. Instead, an explicit string solution dual to the bosonic latitude Wilson loop is not known yet. A preliminary discussion can be found in [62] and steps towards the solution of the problem appeared in [63].

As discussed in [60], classically the latitude fermionic WL is cohomologically equivalent to a linear combination of bosonic latitudes. In fact, one can show that

$$W_F(v) = \frac{e^{-\frac{i\pi v}{2}} W_B(v) - e^{\frac{i\pi v}{2}} \hat{W}_B(v)}{e^{-\frac{i\pi v}{2}} - e^{\frac{i\pi v}{2}}} + \mathcal{Q}(v) - \text{exact term} \tag{39}$$

where $\mathcal{Q}(v)$ is a linear combination of super-Poincaré and superconformal charges preserved by all the operators [60]. We note that for $v = 1$, it reduces to $W_F = \frac{1}{2}(W_B + \hat{W}_B)$, up to $\mathcal{Q}$-exact terms [52].

If this equivalence survives at quantum level, taking the vacuum expectation value of both sides of (39), we can determine $\langle W_F(v) \rangle$ as a linear combination of the bosonic $\langle W_B(v) \rangle$, $\langle \hat{W}_B(v) \rangle$. However, in three dimensions, the evaluation of Wilson loop vev is affected by framing ambiguities [64][11]. Therefore, the problem of understanding how the classical cohomological equivalence gets implemented at quantum level is strictly interconnected with the problem of understanding framing.

This problem was extensively discussed in [68–70], where it was shown that the cohomological equivalence gets enhanced at quantum level in exactly the same form (39) if the vev is computed at *framing* $v$, where $v$ is the latitude[12]. Precisely, if perturbatively we define ($\lambda = N/k$),

$$\langle W_B(v) \rangle_v \equiv e^{i\Phi_B(v,\lambda)} \langle W_B(v) \rangle_0 + O(k^{-3}) \quad , \quad \langle \hat{W}_B(v) \rangle_v \equiv e^{i\hat{\Phi}_B(v,\lambda)} \langle \hat{W}_B(v) \rangle_0 + O(k^{-3})$$
$$\langle W_F(v) \rangle_v \equiv \langle \hat{W}_F(v) \rangle_0 + O(k^{-3}) \tag{40}$$

where $\langle \cdot \rangle_0$ stands for expectation values computed in ordinary perturbation theory with dimensional regularization and $\Phi_B, \hat{\Phi}_B$ are the framing functions, the quantum cohomological equivalence is conjectured to be [60]

$$\langle W_F(v) \rangle_v = \frac{e^{-\frac{i\pi v}{2}} \langle W_B(v) \rangle_v - e^{\frac{i\pi v}{2}} \langle \hat{W}_B(v) \rangle_v}{e^{-\frac{i\pi v}{2}} - e^{\frac{i\pi v}{2}}} \tag{41}$$

This identity was checked perturbatively, up to two loops for $v = 1$ in [71–73], whereas for generic $v$, it was successively tested in [60]. At this order, the framing function is given by $\Phi_B(v, \lambda) = -\hat{\Phi}_B(v, \lambda) = \pi v \lambda + O(\lambda^3)$.

A direct perturbative evaluation of $\langle W_B(v) \rangle_v$ at framing $v$ is done up to three loops, at finite $N$ [4]. Suitably normalizing the operator, the following result is obtained

$$\langle W_B(v) \rangle_v = 1 + i\pi v \frac{N}{k} + \frac{\pi^2}{6k^2}\left(2N^2 + 1\right) + \frac{i\pi^3 N}{6k^3}\left[v^3\left(N^2 + 1\right) + 3v\right] + O\left(k^{-4}\right) \tag{42}$$

whereas $\hat{W}_B$ is simply the hermitian conjugate. Assuming the cohomological identity (41) to be true, one can easily infer the three loop result also for $\langle W_F(v) \rangle_v$.

As discussed in [4], framing seems to have a quite different origin in the undeformed ($v = 1$) and deformed ($v \neq 1$) cases.

For the $v = 1$ Wilson operator, the three-loop result reveals that all the framing effects are encoded into a phase, being the imaginary terms at odd orders associated only to framing dependent Feynman diagrams [68]. Therefore, in this case, Equation (40) holds with no need for $O(k^{-3})$ corrections.

For the latitude instead, an imaginary contribution to $\langle W_B(v) \rangle_v$ arises at three loops, which is framing independent [4]. Therefore, in the general case, the phase in (40) is not entirely due to framing. We should also expect that not all the framing effects are encoded

into a phase, as it happens already at this order for multi-winding Wilson loops [70]. In order to describe the most general situation, it is then convenient to replace Equation (40) as

$$\langle W_B(\nu)\rangle_\nu \equiv e^{i\Phi_B(\nu,\lambda)}\,|\langle W_B(\nu)\rangle|\quad,\quad \langle \hat{W}_B(\nu)\rangle_\nu \equiv e^{i\hat{\Phi}_B(\nu,\lambda)}\,|\langle \hat{W}_B(\nu)\rangle|$$
$$\langle W_F(\nu)\rangle_\nu \equiv |\langle \hat{W}_F(\nu)\rangle| \tag{43}$$

with the understanding that in the no-latitude case, the modulus coincides with the vev evaluated at framing zero, and $\Phi_B, \hat{\Phi}_B$ are the genuine framing functions, whereas for the latitude, this is true only up to two loops.

We note that modding out the framing-zero two-loop result obtained by using ordinary dimensional regularization [60], from result (40), we can infer the expansion of the framing function at this order. In the large $N$ limit, it reads

$$\Phi_B(\nu,\lambda) = -\hat{\Phi}_B(\nu,\lambda) = \pi\nu\lambda - \frac{\pi^3}{6}(\nu^3+2\nu)\lambda^3 + O(\lambda^5) \tag{44}$$

Notably, this expression coincides with the one conjectured in [74], using the relation between circular Wilson loops, Bremsstrahlung functions and the cusp anomalous dimension [28,45,51,60,62,75].

### 4.3. The Matrix Model for BPS Wilson Loops

As reported in Section 2, correlation functions for gauge invariant, BPS operators can be computed, using localization techniques. In particular, this turns out to be true for BPS Wilson loops evaluated on closed paths on $S^3$, as long as they preserve the supercharge used for localizing the functional integral.

The Matrix Models computing the vev of the bosonic 1/6 BPS Wilson loops corresponding to connections (33) and evaluated on the great circle was proposed in [16]. They are simply given by the matrix representation of the partition function $\mathcal{Z}$ in (3) with the following insertions

$$W_B : \; \frac{1}{N}\sum_{a=1}^{N} e^{2\pi\lambda_a} \qquad\qquad \hat{W}_B : \; \frac{1}{N}\sum_{a=1}^{N} e^{2\pi\mu_a} \tag{45}$$

and normalized with the partition function itself. It is important to stress that the matrix model always computes the vevs at framing one since the only point-splitting regularization which does not break supersymmetry on $S^3$ corresponds to taking the original path and the framed one to belong to a Hopf fibration of the sphere.

In principle, prescription (45) provides an exact result for the bosonic operators, which turn out to be complex functions of the coupling and thus, expressible as in (43), with the framing function given by an odd power series in the coupling.

Since the matrix model is invariant under the supercharge that drives the cohomological equivalence between $W_B, \hat{W}_B$ and $W_F$, we immediately obtain

$$\langle W_F\rangle_1 = \frac{\langle W_B\rangle_1 + \langle \hat{W}_B\rangle_1}{2} \tag{46}$$

where the subscript indicates that the results are at framing one. This result turns out to be real, in agreement with (43).

The matrix model can be expanded at small coupling $\lambda$ [16–18], leading to a prediction that can be tested against a genuine perturbative calculation. Indeed, up to three loops, it matches the perturbative result (42) evaluated at $\nu = 1$. The matrix model was also computed at strong coupling, using a Fermi gas approach [76,77]. The leading contribution of $\langle W_F\rangle_1$ at strong coupling matches the exponential behavior predicted from the large $N$ dual description [54]. Matching was found also for the first subleading correction in [78], where the problem of fixing ambiguities in the normalization of the string path integral was reconsidered, and a universal normalization was proposed.

Generalizing the matrix model construction to the evaluation of the latitude Wilson loops is not an easy task, due to the fact that these operators preserve supercharges, which differ from the one used in [16] to localize the path integral and cannot be embedded in the $\mathcal{N} = 2$ superspace formalism easily. However, in [4] a $\nu$-dependent matrix model computing $\langle W_B(\nu) \rangle_\nu$ was proposed, which is a slight deformation of the known one in (45) [16]. The vevs are computed as follows

$$\langle W_B(\nu) \rangle_\nu = \left\langle \frac{1}{N} \sum_{a=1}^N e^{2\pi \sqrt{\nu} \lambda_a} \right\rangle \qquad \langle \hat{W}_B(\nu) \rangle_\nu = \left\langle \frac{1}{N} \sum_{a=1}^N e^{2\pi \sqrt{\nu} \mu_a} \right\rangle \qquad (47)$$

where now the average is evaluated and normalized with the following partition function

$$\mathcal{Z}(\nu) = \int \prod_{a=1}^N d\lambda_a \, e^{i\pi k \lambda_a^2} \prod_{b=1}^N d\mu_b \, e^{-i\pi k \mu_b^2} \qquad (48)$$

$$\times \frac{\displaystyle\prod_{a<b}^N \sinh \sqrt{\nu} \pi (\lambda_a - \lambda_b) \sinh \frac{\pi(\lambda_a - \lambda_b)}{\sqrt{\nu}} \prod_{a<b}^N \sinh \sqrt{\nu} \pi (\mu_a - \mu_b) \sinh \frac{\pi(\mu_a - \mu_b)}{\sqrt{\nu}}}{\displaystyle\prod_{a=1}^N \prod_{b=1}^N \cosh \sqrt{\nu} \pi (\lambda_a - \mu_b) \cosh \frac{\pi(\lambda_a - \mu_b)}{\sqrt{\nu}}}$$

As before, the integral is over a set of $(\lambda_a, \mu_a)$ eigenvalues of the Cartan matrices of $U(N) \times U(N)$.

This matrix model is expected to be the result of localizing the vevs by using the $\nu$-dependent supercharges preserved by $W_B(\nu)$. However, in the absence of a localization procedure that leads directly to (47) and (48), a number of strong consistency checks are available:

(1) First of all, for $\nu = 1$, it reduces to the known matrix model (45) and (3).

(2) A first non-trivial check concerns the partition function (48). Since its value should be independent of the localizing supercharge that we use to infer the matrix model, (48) should provide the ordinary $\nu$-independent partition function of the ABJM model. Indeed, this was successfully checked in [4] where it was shown that expression (48) can be rearranged in such a way that the $\nu$ dependence disappears completely and it ends up coinciding with the ABJM partition function (3). Since such manipulations no longer work when we insert the WL exponentials (47), we correctly expect a non-trivial $\nu$-dependence in the Wilson loop vevs.

(3) Important checks come from comparing the matrix model results at weak and strong couplings with alternative calculations. At weak coupling, its expansion perfectly matches the perturbative result (42). This confirms the intuition that localization should compute Wilson loops at framing $\nu$.

(4) Expressions (47) were computed at large $N$ in the strong coupling limit, using the Fermi gas approach [4]. Applying a genus expansion in powers of the string coupling $g_s = \frac{2\pi i}{k}$, and introducing the new variable $\kappa$ through the identity

$$\lambda = \frac{\log^2 \kappa}{2\pi^2} + \frac{1}{24} + O\left(\kappa^{-2}\right) \qquad (49)$$

the genus-zero terms (that is the leading order in $1/k$) read

$$\langle W_B(\nu) \rangle_\nu \big|_{g=0} = \frac{-\kappa^\nu \, \Gamma\left(\frac{\nu-1}{2}\right) \Gamma\left(\frac{\nu+1}{2}\right) + i \pi \kappa \left(1 + i \tan \frac{\pi\nu}{2}\right) \Gamma(\nu+1)}{4\pi \, \Gamma(\nu+1)} \qquad (50)$$

with $\langle \hat{W}_B(\nu) \rangle$ given simply by the Hermitian conjugate. Using the cohomological equivalence in (39) the strong coupling expansion for the fermionic latitude Wilson loop can be easily inferred to be

$$\langle W_F(\nu) \rangle_\nu \big|_{g=0} = -i \frac{2^{-\nu-2} \kappa^\nu \Gamma\left(-\frac{\nu}{2}\right)}{\sqrt{\pi}\, \Gamma\left(\frac{3}{2} - \frac{\nu}{2}\right)} \tag{51}$$

Remarkably, its leading behavior at strong coupling

$$\langle W_F(\nu) \rangle \sim e^{\pi \nu \sqrt{2\lambda}} \tag{52}$$

reproduces the holographic prediction found in [62]. Moreover, in [79,80] the ratio $\frac{\langle W_F(1) \rangle}{\langle W_F(\nu) \rangle}\Big|_{g=0}$ was computed holographically at strong coupling, at the next-to-leading order, and the result perfectly matches the matrix model prediction (51).

Very recently, the hard task of proving that the matrix model is the result of applying a localization procedure driven by a $\nu$-dependent supercharge was taken on [81]. Though the authors did not manage to solve directly the very difficult problem of bringing the latitude supercharges off-shell as required to make localization work, they managed to show that matrix model (48) emerges as the result of applying the Källén approach [82] (generalized to Chern–Simons theories with matter in [83]) under the assumption that also for the latitude supersymmetry algebra, there exists a kind of topologically twisted Lagrangian as the one considered there, which is on-shell equivalent to the ABJM one.

Before closing this section, we note that the expectation values (47) satisfy the functional identity

$$\partial_\nu \log\left(\langle W_B(\nu) \rangle_\nu + \langle \hat{W}_B(\nu) \rangle_\nu\right) = 0 \tag{53}$$

In other words, the real part of the average $\langle W_B(\nu) \rangle_\nu$ is independent of $\nu$. This non-trivial property is going to be useful for the discussion on the Bremsstrahlung function in the next section.

### 4.4. The Bremsstrahlung Function

In SCFTs, circular Wilson loops have remarkable connections with other physical quantities, such as the Bremsstrahlung function and the cusp anomalous dimension. In this section, we review the main results regarding latitude Wilson loops in the ABJM theory. We will address how this connections have far reaching consequences, primarily because they extend to other physical quantities the possibility of an exact evaluation via localization. Moreover, they assign a privileged role to Wilson operators, which become the meeting point for localization, integrability and conformal bootstrap.

**The definition of B**. The physical definition of the Bremsstrahlung function $B$ is encoded in the expression for the energy $\Delta E$ lost by a massive quark slowly moving in a gauge background with velocity $v$,

$$\Delta E = 2\pi B \int dt\, \dot{v}^2 , \qquad \text{with} \quad |v| \ll 1 \tag{54}$$

In general $B$ is a non-trivial function of the coupling constant of the theory.

In CFTs, it is also related to the cusp anomalous dimension $\Gamma_{cusp}(\phi)$. This is the quantity that weights the singular part of a Wilson operator evaluated on a cusped contour, that is a contour made by two semi-infinite straight lines that meet at a point forming an angle $\phi$. Close to the cusp short distance singularities appear, which exponentiate as

$$\langle W^{\angle} \rangle_\phi \sim e^{-\Gamma_{cusp}(\phi) \log \frac{L}{\epsilon}} \tag{55}$$

Here $L$ is the length of the two straight lines (the IR regulator) and $\epsilon$ the UV regulator. For small angles, $\phi \ll 1$, the cusp anomalous dimension behaves as $\Gamma_{cusp}(\phi) \sim -B\,\phi^2$ [45], where $B$ is the Bremsstrahlung function defined in (54).

In ABJM theory, since there are both bosonic and fermionic Wilson operators we can define different types of cusped operators and consequently different types of Bremsstrahlung functions [51,84].

First, if we compute UV divergent contributions to the fermionic, $1/2$ BPS operator $W_F$ close to a cusp, we obtain

$$\langle W_F^{\angle}(\theta)\rangle_\phi \sim e^{-\Gamma_{cusp}^{1/2}(\phi,\theta)\log\frac{L}{\epsilon}}\ , \qquad\qquad \Gamma_{cusp}^{1/2}(\phi,\theta) \underset{\phi,\theta\ll 1}{\sim} B_{1/2}\left(\theta^2-\phi^2\right) \qquad (56)$$

where $\theta$ is an internal angle that describes possible relative rotations of the matter couplings in the Wilson loops defined on the two edges of the cusp (encoded in the parameter $\nu = \cos\theta$ of Section 4.2). The particular $\theta, \phi$ dependence that appears at small angles in (56) is dictated by the fact that for $\theta^2 = \phi^2$ the cusped operator becomes BPS-thus finite-and the cusp anomalous dimension has to vanish.

Second, if we study the short distance behavior of a $1/6$ BPS bosonic operator $W_B$ near a cusp, no BPS enhancement occurs in this case, and the small angle behavior of the cusp anomalous dimension is given in terms of two different Bremsstrahlung functions as follows

$$\langle W_B^{\angle}(\theta)\rangle_\phi \sim e^{-\Gamma_{cusp}^{1/6}(\phi,\theta)\log\frac{L}{\epsilon}}\ , \qquad\qquad \Gamma_{cusp}^{1/6}(\phi,\theta) \underset{\phi,\theta\ll 1}{\sim} B_{1/6}^{\theta}\,\theta^2 - B_{1/6}^{\phi}\,\phi^2 \qquad (57)$$

Beyond being related to the cusp anomalous dimension, the $B$ functions have also remarkable relations with two-point correlation functions of the one-dimensional defect SCFT defined on a Wilson line. Focusing for instance on the evaluation of $B_{1/2}$, from (56), where we set $\phi = 0$ (infinite straight line) it is easy to see that [28,45]

$$B_{1/2} = -\frac{1}{2}\frac{\partial^2}{\partial\theta^2}\log\langle W_F(\theta)\rangle\Big|_{\theta=0} = \frac{1}{2N}\left(\frac{4\pi^2}{k^2}(c_s+\hat{c}_s) - \frac{\pi}{k}c_f\right) \qquad (58)$$

where $c_s, \hat{c}_s$ and $c_f$ are the coefficients appearing in the two-point functions on the Wilson line (see definition (32)) for dimension-one operators in the displacement multiplet built up from the fundamental ABJM fields in (5)[13]

$$\langle\!\langle (Y_a\bar{Z})(s_1)\,(Z\bar{Y}^b)(s_2)\rangle\!\rangle = \delta_a^b\,\frac{c_s}{(s_1-s_2)^2}\quad,\quad \langle\!\langle (\bar{Z}Y_a)(s_1)\,(\bar{Y}^bZ)(s_2)\rangle\!\rangle = \delta_a^b\,\frac{\hat{c}_s}{(s_1-s_2)^2}$$

$$\langle\!\langle \chi_a^+(s_1)\,\bar{\chi}_+^b(s_2)\rangle\!\rangle = i\delta_a^b\,c_f\,\frac{(s_1-s_2)}{|s_1-s_2|^3} \qquad (59)$$

These results simply follow from the fact that in the Wilson line, the $\theta$ (alias $\nu$) parameter appears inside the couplings to the matter fields. Therefore, deriving, with respect to the parameter, brings down operators in the matter sector.

**The exact prescription for computing $B$.** All the $B$s are, in general, functions of the coupling constant $\lambda = N/k$ of the ABJM theory and require specific determination. Although, in principle, they could be computed directly from the cusp anomalous dimension, this is generally obstructed by the fact that the perturbative evaluation of $\Gamma_{cusp}$ is not an easy task, being already at low orders. A more successful approach could arise if we were able to relate these quantities to physical observables that are exactly computable via localization. The striking result found in [45] by exploiting the line-to-circle mapping in CFTs provides an exact prescription for computing $B$ in four-dimensional $\mathcal{N}=4$ SYM in terms of the $1/2$ BPS circular Wilson loop, which is amenable of matrix model evaluation.

For the ABJM theory, this problem was originally addressed in [51], where the following prescription for computing $B_{1/6}^{\phi}$ in (57) in terms a $m$-winding circular 1/6 BPS bosonic WL was proposed

$$B_{1/6}^{\phi} = \frac{1}{4\pi^2}\, \partial_m \log|\,\langle W_B^m \rangle\,|\,\Big|_{m=1} \tag{60}$$

A similar prescription was later proposed for computing $B_{1/2}$ in (56) [60] and $B_{1/6}^{\theta}$ in (57) [62] in terms of fermionic (34) and bosonic (33) latitude Wilson loops, respectively

$$B_{1/2} = \frac{1}{4\pi^2}\, \partial_\nu \log|\,\langle W_F(\nu) \rangle\,|\,\Big|_{\nu=1} \quad , \qquad B_{1/6}^{\theta} = \frac{1}{4\pi^2}\, \partial_\nu \log|\,\langle W_B(\nu) \rangle\,|\,\Big|_{\nu=1} \tag{61}$$

These identities were proved in [28,62], respectively, by exploiting the relation between the Bremsstrahlung functions and correlation functions in one-dimensional defect CFTs defined on the circular Wilson loops (the analogues of Equations (58) and (59) on the circle). Moreover, the interesting relation

$$B_{1/6}^{\theta} = \frac{1}{2} B_{1/6}^{\phi} \tag{62}$$

was guessed in [85,86] from a four-loop calculation and finally proved in [74], using a superconformal defect approach. We note that, according to identities (60) and (61), this implies a non-trivial relation between the $\nu$-derivative of the latitude $W_B(\nu)$ and the $m$-derivative of the $m$-winding Wilson loop.

Expanding the matrix models at weak coupling, from (61), we can read the first few orders in the perturbative expansion of the Bremsstrahlung functions to be

$$B_{1/2} \underset{\lambda \ll 1}{=} \frac{\lambda}{8} - \frac{\pi^2}{48}\lambda^3 + O(k^{-5})$$

$$B_{1/6}^{\phi} = 2B_{1/6}^{\theta} \underset{\lambda \ll 1}{=} \frac{\lambda^2}{4} - \frac{\pi^2}{4}\lambda^4 + O(k^{-6})$$

Similarly, expanding the matrix models at strong coupling, we obtain

$$B_{1/2} \underset{\lambda \gg 1}{=} \frac{\sqrt{2\lambda}}{4\pi} - \frac{1}{4\pi^2} - \frac{1}{96\pi}\frac{1}{\sqrt{2\lambda}}$$

$$B_{1/6}^{\phi} = 2B_{1/6}^{\theta} \underset{\lambda \gg 1}{=} \frac{\sqrt{2\lambda}}{4\pi} - \frac{1}{4\pi^2} - \frac{1}{96\pi}\frac{1}{\sqrt{2\lambda}} + \left(\frac{1}{4\pi^3} - \frac{5}{96\pi}\right)\frac{1}{\sqrt{2\lambda}} \tag{63}$$

Perturbative checks up to two loops for $B_{1/6}^{\phi}$ and $B_{1/2}$ can be found in [60,84], whereas a similar check for $B_{1/6}^{\theta}$ is given in [60]. A three-loop calculation of $\Gamma_{cusp}$ [87] provides a non-trivial check for $B_{1/2}$ at this order. At strong coupling, $B_{1/2}$ matches the string prediction at next-to-leading order [75,88].

**The $B$ and the framing.** The exact prescriptions in (61) for computing the Bremsstrahlung functions in terms of latitude Wilson loops lead to a new remarkable interpretation of framing in three-dimensional Chern–Simons-matter theories [60,74,87].

To elaborate on this point, we begin by considering the identity in (61) for $B_{1/2}$. We first substitute $\langle W_F(\nu) \rangle$ there with its expression (41) sustained by the cohomological equivalence, and write the bosonic BPS Wilson loops as in (43) in terms of their moduli and phases. Finally, taking the $\nu$-derivative under condition (53) and evaluating the result at $\nu = 1$, we find

$$B_{1/2} = -\frac{i}{8\pi}\frac{\langle W_B \rangle - \langle \hat{W}_B \rangle}{\langle W_B \rangle + \langle \hat{W}_B \rangle} = \frac{1}{8\pi}\tan\Phi_B \tag{64}$$

where $W_B, \hat{W}_B$ are the underformed bosonic 1/6 BPS WLs corresponding to connections (33) and $\Phi_B$ their framing function (43), evaluated at $\nu = 1$. As already discussed, for

$\nu = 1$, this phase contains *all and only* framing contributions. Therefore, the result (64) suggests that framing—which, in topological Chern–Simons theories, corresponds to integer topological invariants and represents a controllable regularization scheme dependence—in non-topological Chern–Simons-matter theories is no longer a number; rather, it is the function that sources the Bremsstrahlung function.

A similar interpretation holds also for the bosonic $B_{1/6}^{\theta}$. In fact, if we elaborate prescription (61), exploiting identity (53), we easily obtain

$$B_{1/6}^{\theta} = \frac{1}{4\pi^2} \tan \Phi_B(\nu) \, \partial_\nu \Phi_B(\nu) \Big|_{\nu=1} \tag{65}$$

where now $\Phi_B(\nu)$ is the generic bosonic phase function at latitude $\nu$ defined in (43). In this case, as already mentioned, it contains *all but not only* framing contributions. This identity is exploited to perform non-trivial checks of the whole construction. In fact, the four-loop calculation of [85,86] for $\Gamma_{cusp}^{1/6}$ allows to determine $B_{1/6}^{\theta}$ up to this order. Using Equation (65), this in turn provides a prediction for the expansion of $\Phi_B(\nu)$ up to $\lambda^3$ [74]. Merging this result with the two-loop calculation of $|\langle W_B(\nu)\rangle|$ [60], one obtains a three-loop expansion for $\langle W_B(\nu)\rangle_\nu$, which coincides with result (42), obtained by a genuine three-loop calculation of $\langle W_B(\nu)\rangle$ done at framing $\nu$ [4], and is remarkably reproduced by the matrix model average (45) expanded at weak coupling.

Finally, exploiting identity (62), we can write the following chain of equalities [74]

$$B_{1/6}^{\theta} = \frac{1}{2} B_{1/6}^{\phi} = \frac{2}{\pi} B_{1/2} \, \partial_\nu \Phi(\nu) \Big|_{\nu=1} \tag{66}$$

which relates all the Bremsstrahlung functions of the ABJM theory. Eventually, they are all determined by the same $\Phi(\nu)$ phase.

**Connection with integrability.** The link between the Bremsstrahlung functions and the circular BPS Wilson loops—eventually, the matrix models—opens a window on the study of the connection between two different exact techniques in quantum field theory: localization and integrability. This is already evident in four dimensions. In fact, in the planar limit of the $\mathcal{N} = 4$ $SU(N)$ SYM theory, the Bremsstrahlung function was obtained by solving a boundary TBA system of integral equations [89–92]. Therefore, exact results obtained using integrability can be matched with the analogues obtained using localization.

The ABJM theory is also known to be integrable in the planar limit [93–99]. A system of TBA equations was proposed, which, however, involves a still unknown function $h(\lambda)$, mastering the dispersion relation of a single magnon moving on a spin chain [94,100,101]. Although expansions of $h(\lambda)$ were found at weak [102,103] and strong [104] coupling, a prescription for determining it exactly is still unknown. A conjecture for its exact expression was provided in [105] by exploiting its relation with another observable, the slope function describing the small spin limit of SL(2) operators, which is amenable of exact evaluation via localization techniques. At weak coupling, this conjecture was tested up to order $\lambda^3$ [102,106]. At strong coupling, it was tested up to two loops in the string sigma model [104,107,108].

Alternatively, it should be possible to find a three-dimensional analogue of the set of TBA integral equations proposed in [89,90] to determine the Bremsstrahlung functions. Having in this calculation $h(\lambda)$ as an input, a direct comparison with our proposal (61) for $B$ would provide, in principle, an all-order definition for $h$. Matching the localization and integrability results would then be crucial for an exact proof of the conjecture in [105]. Some preliminary steps in this direction involve the exact evaluation of the fermionic cusp anomalous dimension in a suitable scaling limit [109].

*4.5. One-Dimensional SCFT on the Wilson Line*

In general, extended operators break (super)symmetries of the bulk theory. However, as already discussed, a BPS Wilson loop preserves a fraction of superconformal charges.

This operator then supports a one-dimensional SCFT, whose excitations are local operators living on the Wilson contour. In other words, a BPS Wilson loop defines a *superconformal defect*, which is entirely specified by the spectrum of local operators and their correlation functions as defined in (32). Superconformal invariance and broken symmetries constrain the correlation functions to satisfy non-trivial Ward identities that can be used to sort out their structure. In principle, the defect SCFT can be solved by applying the bootstrap machinery [110–112] to determine the spectrum of scale dimensions and the operator product expansion coefficients.

In this section, we give just a sketch of some recent progress in the application of SCFT techniques to the study of Wilson defects in the ABJM theory.

First of all, given the rich spectrum of Wilson loops of the ABJM theory, we can classify two main superconformal defects: the *bosonic defect* living on the 1/6 BPS bosonic operator $W_B$, and the *fermionic defect* living on the 1/2 BPS $W_F$. They define a $su(1,1|1)$ and a $su(1,1|3)$ SCFT, respectively. The parametric family of 1/6 BPS Wilson loops introduced in [56,57] and reviewed in Section 4.1 interpolate between 1/2 and 1/6 defect SCFTs, and can be interpreted as exactly marginal deformations of the defect SCFT [63]. More general 1/6 and 1/12 superconformal defects are described by fermionic and bosonic latitudes, respectively.

The Wilson defects were investigated, using standard algebraic approaches. Defect supermultiplets associated to the broken currents, notably, the displacement multiplet for 1/2 BPS defects and the displacement and the R-symmetry multiplets for the 1/6 BPS defect, were constructed, and Ward identities constraining the structure of two- and three-point functions were derived [28,74,113].

Defect correlation functions were investigated in different contexts and with different purposes. In particular, their relation with the matrix model computing the Wilson loop itself was exploited. The main connection comes from the fact that taking derivatives of the expectation value of parametric Wilson loops with respect to the parameters provides integrated correlation functions for local operators on the Wilson contour, which are, in principle, computable exactly if a matrix model description of the vev is available.

As already mentioned—see Equations (58) and (59)—integrated two-point functions of dimension-one operators belonging to the displacement supermultiplet, inserted on the fermionic latitude defect $W_F(\nu)$, are related to derivatives of $W_F(\nu)$ with respect to $\nu$. They were shown to be a key ingredient in the rigorous proof of identity (61) for $B_{1/2}$ [28].

Integrated two-point functions of biscalar, dimension-one local operators inserted on $W_B$ were considered in [62] to prove identity (61) for $B^\theta_{1/6}$ and in [74] to prove relation (62). These are the expectation values of dimension-one operators belonging to the R-symmetry supermultiplet, arising from small deformations of the latitude bosonic Wilson loop $W_B(\nu)$ with respect to the $\nu$ parameter. Contact terms that master the singular behavior of these correlators at coincident points are responsible for the appearance of an imaginary contribution at three loops [4]. This provides an explanation from the defect perspective of the emergence of a framing independent, imaginary contribution to $\langle W_B(\nu) \rangle$ discussed in Section 4.2. Relating defect correlators to derivatives of the latitude Wilson loop allows to conclude that imaginary terms in the latitude deformation arise from an anomalous behavior of the relevant two–point functions on the defect.

The deep connection between three-point functions of dimension-one scalar bilinears and the matrix model computing $\langle W_B(\nu) \rangle$ was extensively discussed in [114] for the $U(N_1) \times U(N_2)$ ABJ theory in the color limit $N_2 \gg N_1 \gg 1$, where a topological sector seems to emerge.

For the fermionic 1/2 BPS defect, four-point functions of local operators belonging to the displacement supermultiplet were computed at strong coupling, up to the first subleading correction, using the analytic bootstrap approach [113]. The insertions have a dual description in terms of fluctuations of the dual fundamental string in $AdS_4 \times \mathbb{CP}^3$ ending on the Wilson contour at the boundary. The bootstrap solution was shown

to be perfectly consistent with the result obtained in the dual theory via AdS$_2$ Witten diagrams [113].

Defect data have relevant implications also for bulk physical quantities, notably, the Bremsstrahlung function. In [51], it was conjectured that the Bremsstrahlung function $B^\phi$, which is related to the derivatives of BPS Wilson loops according to prescription (60), has also a remarkable relation with the one-point function of the stress-energy tensor on the superconformal defect, according to the famous relation $B^\phi = 2h_\omega$, where $h_\omega$ is the coefficient of the one-point function of $T_{\mu\nu}$. In [115], it was argued that $B^\phi$ is also related to one of the leading coefficients of the anomalous dimension of defect operators at large transverse spin. Finally, Ward identities of the defect theory can be used to link the two Bremsstrahlung functions, $B^\phi$ and $B^\theta$ [74], as described in Section 4.4.

## 5. Conclusions and Perspectives

We have reviewed some recent progress in the study of line defects in the three-dimensional $\mathcal{N} = 6$ ABJM theory. In the first part, we have considered kinematical defects, that is, trivial one-dimensional submanifolds, which support a topological sector of the theory. In the second part, we have focused on dynamical defects realized as latitude bosonic and fermionic BPS Wilson operators.

The existence of one-dimensional topological sectors opens the possibility to determine topological correlation functions exactly, as they are related to the derivatives of the mass-deformed matrix model computing the bulk partition function. At the same time, this relation represents a promising way to reconstruct the bulk SCFT from the data of a simpler subsector. While this relation was proved for $\mathcal{N} = 4, 8$ theories, a full proof for the $\mathcal{N} = 6$ ABJM theory is not available yet. Nevertheless, some indirect evidence was already collected by computing topological correlators in the perturbative regime and matching them with the conjectured expression from the matrix model expanded at weak couplings. These findings support the conjecture that also for the ABJM theory, like for the $\mathcal{N} = 4, 8$ cases, the mass-deformed partition function works as the generating functional for (integrated) correlation functions on the line, and should be strictly linked to the functional integral for a topological one-dimensional quantum mechanics governing the topological correlation functions of the full theory. It would be crucial to prove that a topological quantum mechanics could emerge directly from some localization procedure, describing not only the full topological sector, including operators of arbitrary dimensions, but possibly the monopole sector [34].

Generalizing this construction to dynamical defects, it would be interesting to investigate whether a topological sector can be supported also by the 1/2 BPS Wilson line. This requires understanding if and how a dynamical defect allows for the construction of a non-trivial cohomology realized in terms of local operators of the defect theory. If possible, it would represent a direct tool to relate superconformal data of the bulk theory in terms of the defect ones, and vice versa. This is presently under study [32].

We have reviewed a number of remarkable results obtained in the last few years on generalized (latitude) Wilson loops. The main result concerns the proposal for a $\nu$-latitude Matrix Model that computes bosonic Wilson operator averages exactly, at framing $\nu$. Assuming cohomological equivalence to hold at quantum level at framing $\nu$, this also provides the exact result for the fermionic operators. These are new, exact, interpolating functions that allow to test the AdS$_4$/CFT$_3$ correspondence in the large $N$ limit. In particular, the strong coupling expansion of the bosonic latitude constitutes a brand new prediction, begging for a string theory confirmation.

The deep meaning of non-integer framing, its identification with the latitude parameter and its connection with the Bremsstrahlung function should be better investigated, both at perturbative level and at strong coupling in the matrix model formulation. Framing functions at strong coupling should have a corresponding description in the dual string, though in string theory, there is no notion of framing. It is then crucial to develop an interpretation of framing in the string dual description. To the best of our knowledge, the

only available interpretation is through its relation with the Bremsstrahlung function in Equation (64). The evaluation of the cusp in the string dual model [75,88,108] indirectly computes the framing function of the field theory result at strong coupling.

The exact mastery of latitude that Wilson operators has remarkably follows-up for other physical quantities, primarily, the Bremsstrahlung functions and the correlation functions of the defect theory. Since the Bremsstrahlung functions could be alternatively computed by exploiting the exact solvability of the model, matching localization and integrability results would provide a crucial check of the conjecture in [105] for the interpolating function $h(\lambda)$ of the ABJM theory.

Integrated correlation functions of defect operators of the form $m_J^I(\tau)C_I(\tau)\bar{C}^J(\tau)$ can be extracted in principle from derivatives of the bosonic latitude Wilson loop, with respect to the $\nu$ parameter. Knowing the explicit expression of these correlators from the matrix model would provide information on the OPE data of the defect SCFT. This is definitely something that deserves deeper investigation, along the lines of what was done already in four dimensions [116].

Beyond that, there are still quite a lot of important issues that need to be addressed.

First of all, an important question is to understand the relation between the defect theories defined on $W_B(\nu)$ and $W_F(\nu)$. For instance, we should expect the cohomological equivalence in (39) to play a prominent role in relating SCFT data of the two defect theories. It might turn out that one can reconstruct entirely the defect theory on the fermionic Wilson line from the bosonic one.

As reviewed above, the matrix model has a non-trivial dependence on framing, which ultimately equals the deformation parameter $\nu$. Understanding the meaning of framing from the point of view of the defect theory is definitely an interesting question. Moreover, when computing derivatives of the matrix model with respect to $\nu$, framing contributions will appear, which may affect the evaluation of the correlation functions. It would be then interesting to understand how to disentangle framing effects from the defect correlators.

For clarity, we have focused on the ABJM theory. However, most of the results can be easily generalized to the case of the $U(N_1)_k \times U(N_2)_{-k}$ ABJ theory [3]. In particular, the expressions for the latitude Wilson loops are basically the same, except for the overall normalizing factors that will be functions of $N_1$ and $N_2$. A more general matrix model was proposed also for this theory [4]. A difference between the two matrix models emerges in the evaluation of the partition function, which, in the ABJ case, maintains a non-trivial $\nu$-dependence in its phase [4]. The appearance of this phase could be ascribed to a Chern–Simons framing anomaly discussed in [39,117] and leads to the conclusion that the deformation affects the partition function only in its somehow unphysical part, whereas its modulus is $\nu$-independent. However, this point is still not totally clear and deserves further investigation.

Finally, it would be very interesting to generalize the present investigation to dynamical defects in less supersymmetric theories, notably $\mathcal{N} \geq 2$ quiver Chern–Simons-matter theories, where more general classes of latitude Wilson loops were constructed [58,118,119]. For Wilson loops defined on the maximal circle, the two-loop evaluation was performed in [118]. Since the structure of the superconnections is similar to the one for the 1/2 BPS Wilson loop in the ABJM theory, the topologies of the diagrams are exactly the same. The calculation can then be done by easily exploiting the results for the loop integrals already available for ABJM. Latitude Wilson loops in $\mathcal{N} = 4$ quiver theories were constructed in [119], where also a first proposal for a matrix model computing them can be found, as a clever generalization of the matrix model in (47,48). It would be interesting to compute the Bremsstrahlung functions associated to cusped Wilson operators belonging to different classes and see, for instance, whether the $B$ functions corresponding to different classes turn out to be different. Moreover, their potential connection with circular Wilson loops is an important problem to investigate along the lines of what was described in Section 4.4.

**Funding:** This research received no external funding.

**Data Availability Statement:** Not applicable.

**Acknowledgments:** I am grateful to my senior and young collaborators, Marco S. Bianchi, Gaston Giribet, Nicola Gorini, Luca Griguolo, Luigi Guerrini, Matias Leoni, Andrea Mauri, Hao Ouyang, Michelangelo Preti, Domenico Seminara, Paolo Soresina, Jun-Bao Wu and Jiaju Zhang, who contributed substantially to achieve some of the results presented in this review. A special thanks is devoted to Norma Sanchez for her kind invitation to contribute with this review article to the Open Access Special Issue "*Women Physicists in Astrophysics, Cosmology and Particle Physics*", to be published in [Universe] (ISSN 2218-1997, IF 1.752). This work has been partially supported by Università degli studi di Milano-Bicocca, by the Italian Ministero dell'Università e della Ricerca (MUR), and by the Istituto Nazionale di Fisica Nucleare (INFN) through the "Gauge theories, Strings, Supergravity" (GSS) research project.

**Conflicts of Interest:** The author declares no conflict of interest.

## Notes

1　We use conventions of [4], where $D_\mu C_I = \partial_\mu C_I + i A_\mu C_I - i C_I \hat{A}_\mu$, $D_\mu \bar{C}^I = \partial_\mu \bar{C}^I - i \bar{C}^I A_\mu + i \hat{A}_\mu \bar{C}^I$ and similarly for fermions.

2　For a nice introduction to localization, see, for instance, [15]. Briefly, this technique consists of deforming the original functional integral, which evaluates the partition function by shifting $S \to S + t\,QV$, where $Q$ is an odd symmetry generator, satisfying $Q^2 = \delta$, with $\delta$ being a bosonic symmetry, $V$ a positive semi-definite fermionic functional and $t$ a positive number. As long as $\delta V = 0$, it is easy to see that the functional integral does not depend on $t$. Therefore, it can be computed at $t \to +\infty$, where it localizes on the zero locus of $QV$. In a non-abelian gauge theory these are matrices, so that the original functional integration is traded for a finite dimensional matrix integral. In this limit the saddle point approximation becomes exact and the integrand is simply given by the exponential of the classical action evaluated at the saddle points times the one-loop determinant resulting from the integration on the quadratic fluctuations of the fields around their saddle values. The whole procedure requires compactifying the theory on the sphere in order to avoid IR divergences, but if we are dealing with a SCFT, this is not an issue.

3　We use notations and conventions in [26]. In particular, the two superalgebras and their irreducible representations are spelled there, in Appendices B and C.

4　Since the construction is the same for $\mathcal{Q}_+$ and $\mathcal{Q}_-$, we will use the generic symbol $\mathcal{Q}$ to indicate one of the two supercharges.

5　We use the notation $\mathcal{B}^{\frac{1}{N}, \frac{1}{M}}_{m; j_1, j_2}$ to label a short irreducible representation, whose superconformal primary is annihilated by $\frac{1}{N}$ and $\frac{1}{M}$ fractions of $Q$ and $\bar{Q}$ supercharges in (4), respectively.

6　This construction can be extended to Coulomb branch operators [34], complicated by the presence of monopole operators, and to more general manifolds [35].

7　These operators can be constructed from the lowest component of some flavor symmetry multiplet, therefore the *a* index runs from 1 to the dimension of the flavor symmetry algebra.

8　For a comprehensive review on Wilson loops in three-dimensional Chern–Simons-matter theories, we refer the reader to [43].

9　The great circle corresponds to $\theta = 0$.

10　The fermionic couplings correctly reduce to the ones on the great circle on $S^3$ as given in [52].

11　These are finite regularization ambiguities associated to singularities arising when two fields running on the same closed contour clash. In perturbation theory, this phenomenon is ascribable to the use of point–splitting regularization to define propagators at coincident points [65–67].

12　Though in topological Chern-Simons theories framing is an integer [64], in non-topological theories it generalizes to a non-integer number [60]. Therefore, it can no longer be ascribable to ambiguities associated to point-splitting regularization in perturbation theory.

13　Here the plus components of the fermions are defined as $\chi^+ = \frac{1}{\sqrt{2}}(\chi^1 + \chi^2)$, and $\bar{\chi}_+ = \frac{1}{\sqrt{2}}(\bar{\chi}_1 + \bar{\chi}_2)$.

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
