# Peer review of "Superconformal Line Defects in 3D"

_universe, doi:10.3390/universe7090348_

Round 1

Reviewer 1 Report

The manuscript “Superconformal line defects in 3D” by the author Silvia Penati is a review article concerned with the description of line defects in 3D Chern-Simons-matter SCFTs and, the ABJM theory. The article presents both kinematical and dynamical defects. For kinematical defects the article revisits correlation functions and its derivation from the mass-deformed ABJM theory. For dynamical defects the author revisits the theory with operators which are latitude Wilson loops and its relationship with matrix models. The relation among different concepts as latitude Wilson loops, defect SCFT and the Bremsstralung function is also detailed discussed

The paper is well written, the main argumentation can easily be followed, and the results are explained in clear way. Moreover, the author has actively participated of this development as can be seen from the References section. I will be happy to recommend the paper for publication in Universe.   

Author Response

I thank the referee for her/his comments

Reviewer 2 Report

comments and suggestions are in the attached file

Author Response

I am grateful to the referee for her/his comments. I have done the following corrections to answer her/his question about framing and implement her/his suggestion.

  • Question – This is a very interesting question for which we do not have a clear answer yet. However, I have added a new paragraph at page 21, immediately after the paragraph ending with “….begging for a string theory confirmation. “ to address this problem.
  • Suggestion – I have rewritten the last paragraph of the section “Conclusions and perspectives” (page 22), by including more details on what has been already done for N>= 2 theories and what would be interesting to do.

Reviewer 3 Report

Comments are presented in the pdf uploaded below.

Author Response

(The authors gave the same response as above.)
